# Smart Transition to Climate Management of the Green Energy Transmission Chain

Olena Borysiak [1], Tomasz Wołowiec [2], Grzegorz Gliszczyński [3], Vasyl Brych [1] and Oleksandr Dluhopolskyi [2,4,*]

1. Education and Research Institute of Innovation, Environmental Management and Infrastructure, West Ukrainian National University, 46009 Ternopil, Ukraine
2. Institute of Public Administration and Business, University of Economy and Innovation in Lublin, 20-209 Lublin, Poland
3. Faculty of Management, Lublin University of Technology, 20-618 Lublin, Poland
4. Faculty of Economics and Management, West Ukrainian National University, 46009 Ternopil, Ukraine
* Correspondence: dlugopolsky77@gmail.com; Tel.: +380-682750430

**Abstract:** Climate challenges in recent decades have forced a change in attitude towards forms of environmental interaction. The International Climate Conference COP26 evidences the relevance of the climate issue at the global level in Glasgow (November 2021). A decrease in natural energy resources leads to a search for alternative energy sources. Given this, this article is devoted to studying the peculiarities of the transition to climate management of the green energy transmission chain based on the circular economy and smart technologies. This paper has used simulation modeling to develop an algorithm for applying a smart approach to climate management of the green energy transmission chain based on the work of Industry 4.0 technologies. The result of this modeling will be the importance of strengthening the ability to develop intersectoral partnerships to create climate-energy clusters based on a closed cycle of using energy resources and developing smart technologies. At the same time, it has been found that COVID-19 has changed the behaviour of energy consumers towards the transition to the use of energy from renewable sources that are carbon neutral. With this in mind, this article has assessed the climate capacity of industries to use green energy from renewable sources based on resource conservation (rational use of energy resources) and climate neutrality. The industries of Ukraine, which are the largest consumers of energy and, at the same time, significantly affected by climate change, were taken for the study: industry, transport, and agriculture. The methodology for determining the indicator of the climate capacity of sectors in the transition to green energy has been based on the correlation index (ratio) of the consumption indicator of various types of energy by industries (petroleum products; natural gas; biofuels and waste; electricity) and the indicator of gross value added of industries in pre-COVID-19 and COVID-19 conditions. The results have indicated that the use of energy from renewable sources (biofuels and waste) for the production of goods and services, as well as the economical nature of the provision of raw materials (biomass and faeces) are factors that ensure climate industry neutrality and enhance its climate capability. The prospects of such effects of assessing the climate capacity of sectors will be the basis for the rationale to develop intersectoral partnerships to create climate-energy clusters based on a closed cycle of using energy resources and developing smart technologies.

**Keywords:** climate change adaptation; decarbonization; energy security; renewable energy sources; energy market; climate innovation; smart technologies; industry 4.0; circular economy

## 1. Introduction

The consequences of climate change lead to the search for innovative approaches to the economical use of natural resources by strengthening energy security. The current European action is to develop measures to adapt to climate change by 2030 and 2050 and disseminate them through the formation of an extensive regulatory framework (for example, the European Green Deal (2019) [1], thus forging a climate-resilient Europe Climate Change

(2021) [2]). Their principles are implemented by developing political and legal instruments of the national climate policy (Nationally determined contributions, Long-term strategies for low-carbon development, National plans for energy and climate, and National strategy for adaptation to climate change). The EU is committed to climate neutrality by 2050 and has a more ambitious goal of reducing emissions by at least 55% by 2030 compared to 1990.

The energy sector is one of the industries whose activities negatively impact the climate [3]. In recent years, natural gas has been widely used as a primary clean energy source to replace coal, aiming to reduce the severe environmental pollution caused by coal-fired district heating in winter [4]. The rapid growth in natural gas consumption has placed a significant strain on natural gas production and transportation, which has affected residents' regular demand for heating. Therefore, a district heating (DH) system must accurately forecast natural gas consumption. In [5], a long-term price guidance mechanism for flexible energy service providers is considered which is based on stochastic differential methods that mobilize energy flexibility by indirectly controlling demand for flexible energy systems using reasonable price signals.

Recently, the European Union has tightened the targets set to reduce carbon emissions. The energy production sector, particularly the district heating system, is still dominated by the combustion of fossil fuels, which significantly contributes to such emissions [6,7]. At the same time, one of the most sustainable solutions for heating buildings in Europe is district heating, highlighting the need to integrate renewable energy sources into the heating supply. While district heating is vital to a sustainable future, it requires extensive planning and long-term investment [8,9].

According to the International Energy Agency, energy efficiency (40%) and renewable energy (30%) will play a critical role in preventing global temperatures from rising more than 2 °C and reducing $CO_2$ emissions by 2050 [10]. Renewable energy is critical in decarbonizing the electrical system and mitigating the effects of anthropogenic climate change. However, renewable energy accounts for no more than 25% of the world's generating capacity, with 16% hydropower and about 5% solar (SPP) and wind (WPP) power plants. Hydropower is vulnerable to changes in river water levels and temperature due to global warming [11].

It was predicted that district power supply systems will move to low-temperature district heating and high-temperature district cooling [12]. G. Quirosa, M. Torres, V. Soltero, and R. Chacartegui [13] investigated the application of ultra-low temperature district heating and cooling systems with operating temperatures from 6 to 40 °C to integrate renewable sources with a storage strategy using the distribution network as a storage system.

At the same time, it should be noted that district heating and cooling networks connect and distribute thermal energy resources in the network by sources and needs. As a result, ensuring the optimal distribution of thermal resources in a spatially distributed network and creating carbon-neutral energy systems plays a unique role. X. Li, A. Walch, S. Yilmaz, M. Patel, and J. Chambers [14] proposed a spatial clustering method, transportation theory, and linear programming to maximize distributed resources under spatial constraints, allowing large-scale analysis of a wide range of geospatially-constrained resources, especially when applying renewable energy mapping to supply district heating and cooling.

Introducing smart city and climate-neutral technologies in energy security is particularly essential [15–17]. In addition, the European Energy Efficiency Directive (EED 2012/2018) obliges the Member States to have all electricity meters remotely accessible for reading until January 2027 [18]. M. Tunzi, and S. Svendsen [19] proposed a plan for a provincial integrated energy services platform based on SCADA. The platform is based on CPS and uses the development of intelligent interactive and business management applications as the main line of communication with electricity consumers, energy service providers, government departments and other parties. In turn, using the EnergyPlan platform [20] makes it possible to simulate the operation of intelligent energy systems using renewable energy sources.

For reliable heat supply and cooling, the use of a geographic information system (GIS) identifies heat sources that can be used to provide heat or remove excess heat. H. Pieper, K. Lepiksaar, and A. Volkova [21] proposed a method for identifying possible heat sources for large heat pumps and chillers, combining geospatial data on administrative units, industrial facilities, and natural water bodies.

In the context of the study of the peculiarities of the transition to climate management of the green energy transmission chain and the integration of smart technologies in energy, the importance will be the use of a smart approach to climate management of the green energy transmission chain based on intersectoral cooperation and the circular use of energy from renewable sources.

## 2. Methods

Combining sectors is necessary to effectively integrate renewable energy sources since almost all renewable energy sources depend on variations in environmental parameters [13]. In this context, an important role is played by the development of an algorithm for applying the smart approach to climate management of the green energy transmission chain based on the circular interaction of enterprises in the agricultural, energy, and transport industries. The concept of this study is that the production and transition to the recycling of agro-bioresources is both a way to neutralize the negative impact on the climate (growing photosynthetic plants) and an alternative source of energy (biofuels). It has been established that, in thermal power engineering, the use of renewable energy sources (biofuels) and waste energy obtained from the transport and housing and communal (household) sectors is regarded as a way to transition to climate-neutral thermal energy production. In particular, in thermal power plants, the transfer of such waste energy occurs based on reverse logistics (recycling of energy resources).

To achieve the established purpose and solve certain items, we used general scientific and specific methods, such as content analysis, system analysis, expert assessments, economic-statistical and comparative analysis, simulation modelling, graphical and tabular presentation, and abstract logical ways.

The idea of substantiation of transition to climate management of the green energy supply chain by clustering enterprises in the agricultural, energy, energy service, transport and housing and communal (household) sectors based on the circular economy is justified by a combination of systemic and synergistic approaches to achieve the goal of the research, which made it possible to determine peculiarities of the transition to climate management of the "green" energy transmission chain. The methodological novelty of this approach is the possibility of developing a unified system of models within the framework of intersectoral cooperation, aimed at optimizing the development of "green" agriculture and energy by optimizing the processes of production, supply, and consumption of plant bioresources (agro-raw materials), the transition to the basics of a circular economy, the safety of the population, and the development of "green" transport.

The hypothesis was formed based on the results of our previous studies to determine the factors of influence on the interaction of agricultural enterprises and enterprises for the production of "green" energy to optimize the biomass supply chain [22], model the communication algorithm of energy service companies, transport users in the transition to "green" energy consumption [23], and use optimization methods and models to study the benefits of the transition to "green" energy (maximizing the decarbonization of energy and minimizing the cost of energy consumption) [24,25].

Highlighting the trend of integrating smart technologies into the energy sector was the basis for studying the peculiarities of a smart transition to climate management of the green energy transmission chain. In this study, simulation modeling made it possible to develop an algorithm for applying a smart approach to climate management of the green energy transmission chain based on the work of Industry 4.0 technologies.

It should be noted that a distinctive feature of forming a climate-neutral green energy supply chain is the provision of a closed cycle of energy-resource consumption (primary

and secondary energy) and, as a result, the achievement of a zero-carbon footprint in climate change mitigation measures. In such a chain, we have proposed to single out the subjects of production and supply of primary energy (agricultural enterprises as producers of biomass, i.e., energy plants), processing of primary energy into secondary energy (enterprises to produce biofuels and enterprises of "green" thermal power industry), and supply and service of secondary energy (distribution stations, energy service companies) directly to consumers (households, transport sector).

The innovativeness of building such a chain was in the observance of the climate neutrality principle at all stages of energy conversion, which involves laying the basis for the management model for the provision of "green" energy services to obtain such an optimization effect as maximizing the environmental effect (decarbonization of the environment) and minimizing the cost of energy consumption [24]:

$$f(P) = \sum_{i=1}^{n} p_i t_i \rightarrow \max \tag{1}$$

$$f(V) = \sum_{j=1}^{m} v_j c_j \rightarrow \min \tag{2}$$

where $f(P)$ *i* $f(V)$—functions of maximizing the environmental effect from the provision of "green" energy services $t_i$ and minimizing the cost of energy consumption $c_j$; $p_i$—an indicator of the decarbonization level for the use of "green" energy services $t_i$; $v_j$—consumption costs of the corresponding type of energy $c_j$; $n$—the number of types of "green" energy services ($i = 1 \ldots n$); $m$—the number of energy types ($j = 1 \ldots m$).

Applying simulation modeling in Figure 1 showed an algorithm for a smart transition to climate management of the green energy transmission chain based on intersectoral interaction through a closed cycle of energy resources use, which involves the creation of regional climate-energy clusters.

In order to optimize costs and increase the efficiency of the participants in such a chain (climatic energy cluster), the following functions are provided:

- smart industry centers for managing climate capacity (maybe at the level of one enterprise or group of enterprises in the industry) responsible for monitoring the economic and environmental efficiency of production; the use of energy resources (agricultural smart climate capacity management center); bioprocessing (bioenergy, defense, food); smart-climatic capacity management center, "green" energy (electricity/heat) climate capacity management smart center; a consumer (household, transport, industrial) climate capacity management smart center;
- intersectoral smart centers for climate interaction responsible for optimizing risk factors in the transfer of energy resources;
- energy service smart centers, which are responsible for supporting the adoption of energy-efficient decisions at the industry level (individual enterprise) and the introduction of climate-neutral technologies in the energy chain;
- smart reverse logistics center, which is responsible for processing waste (reverse logistics) from the main activity of consumers of energy resources (for example, the use of waste heat from transport activities to produce thermal energy).

At the same time, the global community is currently underprepared for the growing intensity, frequency, and prevalence of climate change impacts, significantly as emissions rise. Climate resilience needs to be built quickly—moving from heightened public awareness and concern to massive adaptation action. Applying innovative measures for intersectoral interaction to enhance environmental and energy security and transition to climate-controlled energy resource management based on the circular economy and Industry 4.0 technologies is particularly important.

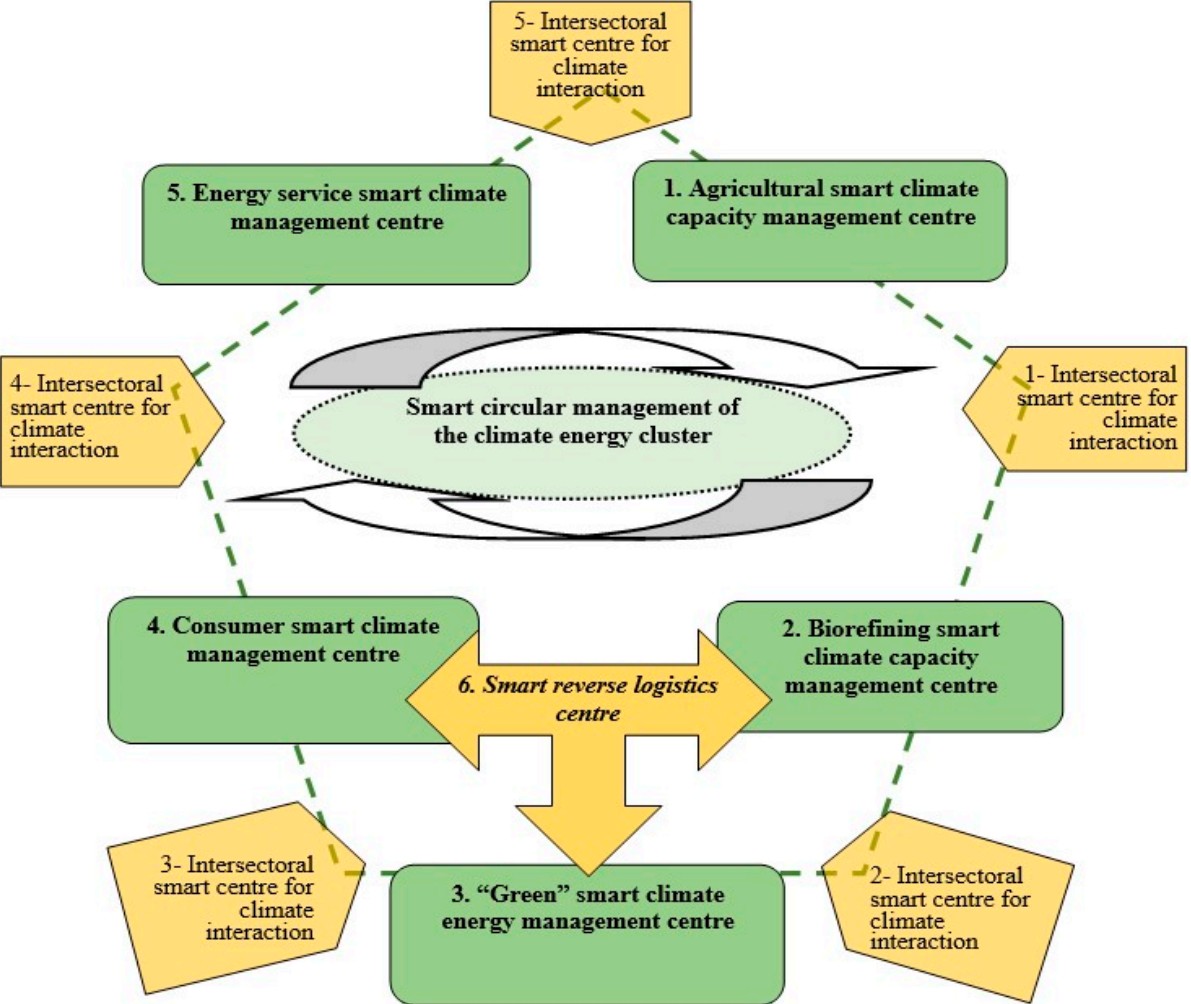

**Figure 1.** Smart Approach to Climate Management of the Green Energy Transmission Chain. Source: author's development.

Given this, we have proposed to assess the climate capacity of industries that are the largest consumers of energy and, at the same time, are affected by climate change transition (for example, industry, transport, and agriculture in Ukraine) to green energy based on resource conservation (rational use of energy resources) and climate neutrality [26–28]. At the same time, it should be noted that COVID-19 [29] has changed the behaviour of energy consumers towards the transition to the use of energy from renewable sources, which is carbon neutral.

In the value structure of final products (output), energy use costs are classified as intermediate costs, while gross value added is defined as the difference between output and intermediate consumption. As a result, we have based the methodology for determining the indicator of the climate capacity of industries in the transition to green energy ($I_{CCi}$) on the correlation index (ratio) of the indicator of consumption of various types of energy (petroleum products; natural gas; biofuels and waste; electricity) by industries ($C_i$) and the indicator of the gross additive of industries ($A_i$) in pre-COVID-19 and COVID-19 conditions (Table 1):

$$I_{CCi} = \sum_{i=1}^{n} R_i \times \left( \frac{\sum_{i=1}^{n} C_i}{A_i} \right), \tag{3}$$

where $I_{CCi}$—the index of climate ability of transition to green energy $i, c, \ldots n + 1$- branches in pre-COVID-19 and COVID-19 conditions; $A_i$—the value of the gross value added indicator of the *i*-sector (industry; transport; agriculture); $C_i$—the value of the indicator

of energy consumption by its types (petroleum products; natural gas; biofuels and waste; electricity) *i*-sectors; $R_i$—the value of the indicator of energy consumption from renewable sources (biofuel and waste, electricity) of the *i*-sector.

**Table 1.** The indicator of the climate capacity of Ukrainian industries in the transition to green energy in pre-COVID-19 and COVID-19 conditions.

|  | Industry | Transport | Agriculture |
|---|---|---|---|
| 2016 | 5.8 | 2.2 | 0.1 |
| 2017 | 4.4 | 2.3 | 0.2 |
| 2018 | 2.1 | 1.5 | 0.2 |
| 2019 | 5.4 | 3.3 | 0.2 |

Source: author's development.

It was diagnosed that the use of natural energy sources is a factor that leads to an increase in the consumer cost of the finished product (service), which is accompanied by an increase in the energy intensity of the industry and leads to a decrease in gross value added. The use of such types of energy as biofuels and waste for the production of goods and services, given the economical nature of the provision of raw materials (biomass and waste), is a factor that ensures climate industry neutrality and enhances its climate capability. Such results of the empirical study are the basis for substantiating the feasibility of justifying the development of intersectoral partnerships to create climate-energy clusters based on a closed cycle of using energy resources and the development of smart technologies.

## 3. Results and Discussions

### 3.1. Cross-Sectoral Approach to Climate Management of the Green Energy Supply Chain

Energy production is one of the sources of greenhouse gas emissions. At the same time, seasonal decrease or increase in demand for energy resources at the consumer level due to climate change negatively affects the balancing of the energy system. Among the priority measures for adaptation to climate change, one can single out the reconstruction and repair of old hydroelectric power plants to increase their resilience to climate change and improve safety to protect hydropower systems from climate change.

One of the elements of energy adaptation to climate change is the so-called "green transition", when the state begins to receive the bulk of the energy from renewable sources. The energy strategy of Ukraine until 2035 provides for the replacement of coal and oil with natural gas and renewable energy sources (Table 2). Natural gas, nuclear energy and renewables are predicted to provide 80% of primary energy in 2035 [30]. In addition, overcoming the consequences of the impact of COVID-19 is on the energy security agenda of the European Union and Ukraine. In April 2020, the International Energy Agency prepared the report "Global Energy Review 2020. The Impact of the COVID-19 Pandemic on Global Energy Consumption and CO2 Emissions" [31], in which was noted the overall decline in energy consumption, especially from traditional sources. It is predicted that renewable energy sources with zero carbon impact on climate change can meet 80% of electricity demand growth over the next ten years. By 2025, renewable energy sources will replace coal as the primary means of generating electricity [31].

This change in behavior and energy consumption during the pandemic has been directly reflected in global levels of $CO_2$ emissions. In 2020, an overall reduction of 2.58 Gt of energy-related $CO_2$ emissions was recorded. Overall, $CO_2$ emissions decreased by an average of 26% (at peaks) during the broad lockdown in selected countries.

**Table 2.** Structure of energy use in Ukraine, %.

| Type of Energy | 2020 | 2035 | Deviation 2035 from 2020 |
|---|---|---|---|
| Natural gas | 32 | 30 | −2 |
| Nuclear energy | 20 | 25 | +5 |
| Renewable energy sources (wind power stations, solar power stations, biomass, thermal energy) | 4 | 25 | +21 |
| Coal | 30 | 13 | −17 |
| Oil | 14 | 7 | −7 |

Source: adapted by the authors with permission from [30].

Because of the need to decarbonize the energy sector, forming an algorithm for the transition to climate management of the "green" energy transmission chain is particularly important in the energy market. In general, the transition to a climate-neutral energy development involves the development of measures to improve the overall energy efficiency of the entire energy transformation chain by increasing the efficiency of energy production and supply (considering the transformation chain from resource to final energy) and increasing the efficiency of energy consumption (considering energy conversion) [32–34]. This can be achieved with a comprehensive analysis of multi-energy systems, namely the entire chain of energy conversion from a source of energy resources (for example, natural resources) and their production (extraction of natural resources as primary energy) until it is converted into an appropriate type of energy for consumption (heat energy, electricity as secondary energy), the distribution of such energy (in the form of light, heat as useful energy) and the direct provision by the relevant energy service (passive systems that do not convert energy) of energy services to consumers.

It should be noted that, according to forecast data [30], by 2030, the structure of consumption of energy types by the source of coordination will change (Table 3) in the direction of preference for alternative types of energy (renewable energy sources—from 10% to 22%; biomass—from 4% to 7%). It is predicted that along with the developed types of renewable energy sources (solar energy, wind energy, hydropower, geothermal energy, solid biofuels, biogas), other sources (hydrogen, tidal energy, current energy, thermal energy of the ocean) will be used in the future.

**Table 3.** Forecast of world consumption of various types of primary energy in 2030, %.

| Type of Energy | 2019 | 2030 | |
|---|---|---|---|
| | | Under Current Regulations | Sustainable Development Scenario (Subject to the Implementation of Policies that Contribute to the Achievement of the UN Sustainable Development Goals) |
| Oil | 32 | 30 | 29 |
| Coal | 26 | 22 | 17 |
| Natural gas | 23 | 24 | 25 |
| Renewable energy sources (wind and solar power stations) | 10 | 15 | 22 |
| Nuclear Energy | 5 | 5 | 7 |
| Biomass | 4 | 4 | - |

Source: adapted by the authors with permission from [30].

At the same time, we noted that the condition for the functioning of the energy system is the uninterrupted process of energy supply to the network (ensuring a balance

between production and energy consumption in real-time), which complicates the process of switching to the use of energy from renewable sources (peak production at daytime hours 9.00–18.00, while consumption is in the evening hours 18.00–22.00). After all, it is impossible to regulate the operation of wind and solar power stations, so it is necessary to turn on/off balancing capacities and limit the operation of renewable energy sources in case of their exhaustion. Given the growing trend of such stations in Ukraine from 2 in 2019 to 20 in 2020, integrating their capacities into the network is becoming more complicated, necessitating the introduction of storage capacities and new highly maneuverable capacities [30].

Taking this into account, when developing an algorithm for the transition to climate management of the "green" energy transmission chain, we consider it appropriate to take into account the aspect of developing and implementing a "green" agricultural policy (the use of biomass—the production of energy crops that absorb carbon dioxide) in the context of environmental and energy security. In the context of this issue, the principle of functioning of the circular economy deserves attention, which provides for the waste-free use of available resources from production to consumption, including the reuse of resources on the principle of reverse logistics.

Within forming a green energy transmission chain, we consider transport as both an energy consumer (for example, electric vehicles) and a source to produce this energy due to recycling (reverse logistics) heat generated during transport operation. This waste heat can be used in district heating systems. As a result, reverse waste logistics for the transport sector makes it possible to integrate transport as consumers and producers of "green" thermal energy based on a closed cycle of energy resource consumption.

Thus, the main subjects of climate management in the "green" energy transmission chain based on a closed cycle of energy resources" are agricultural enterprises (producers of biomass, energy crops); enterprises for processing biomass; enterprises for the production of "green" electricity/heat energy; and enterprises of the transport sphere. The end-users of "green" energy are households and business entities. At the same time, we propose to single out energy service companies in this chain as intermediaries in ensuring the optimization of costs for servicing the green energy transmission chain by introducing climate-neutral and energy-efficient technologies. For the effective functioning of such an intersectoral climate-neutral "green" energy transmission chain, it is necessary to develop an appropriate management mechanism for climate policy development in the energy market.

*3.2. Innovative Solutions to Ensure Climate Policy in the Energy Market Based on the Development of Industry 4.0 Technologies*

The definition of the climate management modeling format for the green energy transmission chain is influenced by the development trends of artificial intelligence (smart technologies and industry 4.0), which is accompanied by the transformation of the energy sector and the development of smart energy networks. In general, the operation content of such networks is to ensure the automation of energy distribution, management of technologies in the energy supply chain, optimization of the pricing policy formation system, and consumer feedback. At the same time, among the difficulties of integrating renewable energy sources into energy networks, unstable production and the difficulty of predicting production (depending on the natural and climatic features of the territories) are singled out [28,29].

The algorithm for smart circular management of a climate-energy cluster (Figure 1) is based on the specifics of the Industry 4.0 smart technologies, which make it possible to ensure optimized, integrated decision-making and omnichannel interaction of all participants in the green energy transmission chain. In particular, introducing low-temperature thermal solutions in district heating involves refurbishing buildings and introducing energy-efficient technologies. The implementation of this task leads to the establishment of interaction between households, energy service companies, and producers of "green" thermal energy. The intersectoral smart center will play this role in our climate-energy cluster for climate interaction.

In addition, we will lay the peculiarities of the HEAT 4.0 project as the basis for smart interaction between green heat energy producers and consumers [35–38], which integrates intelligent IT solutions into a new digital framework to achieve a holistic approach to district heating. HEAT 4.0 caters to the digital needs of the entire sector, from production site to distribution to end-users, creating synergies between the design, operation, maintenance, and supply of district heating. Such solutions are called Cross System Services (CSS) and are based on collaboration between component suppliers, university scientists, district heating companies, consultants, and a typical platform provider [36].

At the same time, in order to improve the European integration processes in Ukraine and transnational cooperation in adapting to climate change in the energy sector, a transition to the digitalization of the energy infrastructure, the development of "smart" energy networks, and the promotion of environmental and energy literacy of the population are required. The urgency of solving this problem is complemented by the fact that by 2025 the reform of the energy complex of Ukraine should be completed, priority targets for security and energy efficiency should be achieved, and its innovative renewal and integration with the EU energy sector should be ensured. The solution to this complex problem is impossible without a profound transformation of the entire energy sector of the economy based on sustainable development and strengthening cooperation between Ukraine and the European Union. In particular, this implies the need to develop a strategy for the digital development of "green" energy, tools for building an innovative business model of "green" energy based on digital technologies, methods for assessing the level of digital transformation of "green" energy by analyzing the structures of "green" energy by the criterion of the degree of digital transformation, and organizational support for the digitalization of green energy structures.

Implementing these tasks will improve Ukraine's cross-border cooperation with the countries of the European Union on the application of an innovative approach to achieve optimal energy security and energy efficiency and the implementation of joint measures to adapt to climate change. In addition, this will contribute to the intensive attraction of investments in the renewable energy sector, the development of distributed generation (in particular, the development and implementation of a plan for the implementation of "smart" energy networks (Smart Grids)), and the creation of extensive electric transport infrastructure in the cross-border zone.

## 4. Conclusions

The sustainability of the proposed smart approach to the functioning of the climate-energy cluster is to create conditions for determining and implementing the innovative resource potential of industries (enterprises) to make economically sound decisions in the transition to climate-neutral and economical consumption of energy resources based on the work of smart technologies. The prospects for the implementation of the algorithm for the transition to climate-controlled energy supply are in the fact that it includes the development of partnerships between agricultural enterprises, green energy enterprises, and green transport organizations based on a closed cycle of using industrial crops as a source of decarbonization, energy or raw materials for the development of green transport through the introduction of climate-neutral innovations in the energy sector based on the work of Industry 4.0 technologies.

This provision is based on the idea that the transition to climate neutrality of energy development by 2050 by the European Union and Ukraine provides the implementation of transnational measures to diversify the sources of obtaining and using "green" energy by enterprises and households, to optimize business processes for the production of "green" energy, and to ensure integrated communication of "green" energy producers, local governments, energy service companies and consumers.

In this context, one should take into account the fact that the use of digital technologies in various areas of life contributes to the inclusive development of the economy, the improvement of the level and quality of life of the population, the availability and ratio-

nal use of resources, ensured security, and automated processing of large databases. In addition, in the context of decentralization, the role of local governments as facilitators in forming a social network to ensure access to "green" energy through the transition to the smart specialization of the energy sector at the regional level is increasing. Today, economic growth, social justice and environmental protection are interrelated components of cyclic socio-economic development. The prospect of developing a unified system of models within the framework of cross-border and transnational cooperation will be aimed at optimizing the development of "green" energy through the digitalization of energy infrastructure, the development of "smart" energy networks, the improvement of energy efficiency, and the increase of energy and environmental security of the population.

Because of this, the prospects for further research are to determine indicators and assess the vulnerability of participants in the climate-energy cluster to climate change, to test the effectiveness of integrating climate-neutral solutions for the introduction of smart technologies in various industries in order to optimize the use of energy resources, and to develop a methodology for calculating the financial and economic benefits of a smart transition to climate management of the green energy supply chain.

**Author Contributions:** Conceptualization, O.B. and V.B.; methodology, O.D.; software, T.W.; validation, T.W., G.G. and O.B.; formal analysis, O.B. and O.D.; investigation, O.B. and V.B.; resources, O.D.; data curation, G.G.; writing—original draft preparation, O.B., T.W. and G.G.; writing—review and editing, V.B. and O.D.; visualization, O.B.; supervision, O.D.; project administration, V.B. and T.W.; funding acquisition, G.G. All authors have read and agreed to the published version of the manuscript.

**Funding:** This research received no external funding.

**Institutional Review Board Statement:** Not applicable.

**Informed Consent Statement:** Not applicable.

**Data Availability Statement:** Not applicable.

**Conflicts of Interest:** The authors declare no conflict of interest.

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
