# Peer review of "Smart Transition to Climate Management of the Green Energy Transmission Chain"

_sustainability, doi:10.3390/su141811449_

Round 1
Reviewer 1 Report
I read the whole manuscript carefully, and find it was not written well, so this manuscript is suggested to made the following major modifications:
1. There are some typos in the manuscript and the English in the manuscript should be polished.
2. The smart transition to climate management used in this manuscript shoule be introduced in the part of "Induction", and some researched papers related to this method should also be added.
3. The results in the part 4 are not clearly and the authors are suggested to give more details, meanwhile, the datas in this part should be tested to credible.
4. In the part of "Discussion", the authors shoule add some comparisons before and after using this method.
Author Response
Point 1: There are some typos in the manuscript and the English in the manuscript should be polished
Response 1: Edited. (in green)
Point 2: The smart transition to climate management used in this manuscript shoule be introduced in the part of "Induction", and some researched papers related to this method should also be added.
Response 2: Introduced in the parts of "Introduction" and "Methods" (in yellow)
Point 3: The results in the part 4 are not clearly and the authors are suggested to give more details, meanwhile, the datas in this part should be tested to credible.
Response 3: Edited. (in yellow)
Point 4: Extensive editing of English language and style required. Are all the cited references relevant to the research?
Response 4: Edited.
Reviewer 2 Report
1. The Abstract should contain answers to the following questions: What problem was studied and why is it important? What are the important results? What conclusions can be drawn from the results? What is the novelty of the work and where does it go beyond previous efforts in the literature?
2. An updated and complete literature review should be conducted and should appear as part of the Introduction.
3. The summary part of this proposed study should be written more comprehensively. For example, which method was used, what are the parameters examined, it should contain statements containing the results obtained.
4. In the introduction section, the literature studies should be rewritten and elaborated. Further information should be given about the literature studies and its obtained results.
5. There are a lot of long sentences in the manuscript, it is suggested to split them into shorter ones for better understanding.
Author Response
Point 1: The Abstract should contain answers to the following questions: What problem was studied and why is it important? What are the important results? What conclusions can be drawn from the results? What is the novelty of the work and where does it go beyond previous efforts in the literature?
Response 1: Edited. (in yellow)
Point 2: An updated and complete literature review should be conducted and should appear as part of the Introduction.
Response 2: Edited. (in yellow)
Point 3: The summary part of this proposed study should be written more comprehensively. For example, which method was used, what are the parameters examined, it should contain statements containing the results obtained.
Response 3: Edited. (in yellow)
Point 4: In the introduction section, the literature studies should be rewritten and elaborated. Further information should be given about the literature studies and its obtained results
Response 4: Edited. (in yellow)
Reviewer 3 Report
The uploaded material is chaotic and very vague. The content of the article does not indicate the legitimacy of developing the algorithm or the methodology of its development. Practically, the article does not contain an unambiguous algorithm that is to be the result of the Authors' analytical work. Fig. 1 is an illustrative drawing, not described in the content of the article and very little contributing and understandable. The validity of presenting the data contained in Table 1 concerning Ukraine and the forecasts presented in Table 2 in the context of the article has not been proved either. I am asking the Authors to re-edit this article, unambiguously presenting:
1. A research (analytical) problem with an indication and justification of the need to solve this problem.
2. The current state of knowledge in the field of the discussed problem based on bibliographic sources.
3. Methodology of the conducted research and analytical process, taking into account all boundary conditions.
4. The developed algorithm in its full form, based on the presented methodology.
5. Specific and unambiguous conclusions presenting the practical possibilities of using the developed algorithm in environmental as well as technical, economic and social aspects.
The article in the current form presented by the Authors is not a scientific article and is not suitable for publication. Major improvement needed to rework the article.
Author Response
Point 1: The uploaded material is chaotic and very vague. The content of the article does not indicate the legitimacy of developing the algorithm or the methodology of its development. Practically, the article does not contain an unambiguous algorithm that is to be the result of the Authors' analytical work. Fig. 1 is an illustrative drawing, not described in the content of the article and very little contributing and understandable. The validity of presenting the data contained in Table 1 concerning Ukraine and the forecasts presented in Table 2 in the context of the article has not been proved either.
Response 1: The article was edited in the parts of "Abstract", "Introduction", "Methods", "Results and Discussions", "Conclusions", "References"
Point 2: A research (analytical) problem with an indication and justification of the need to solve this problem.
Response 2: Edited. (in yellow)
Point 3: The current state of knowledge in the field of the discussed problem based on bibliographic sources. Methodology of the conducted research and analytical process, taking into account all boundary conditions. The developed algorithm in its full form, based on the presented methodology.
Response 3: Edited. (in yellow)
Point 4: Specific and unambiguous conclusions presenting the practical possibilities of using the developed algorithm in environmental as well as technical, economic and social aspects. Major improvement needed to rework the article.
Response 4: Edited.
Reviewer 4 Report
The contents of the paper are interesting but the entire paper seems to be a sort of introduction to the matter. The simulation model is described but deeper details on possible applications are missing.
The overall impression is of a very elementary dissertation, that needs a deeper and more extensive analysis to be published.
Moreover, even the English language used to present them is quite elementary and confused:
Here some suggestions follow.
Line 115: maybe you intended “Energy production” not “Energy”.
Lines 115-117: the phrase “At the same time …… consumption of energy” is not clear, please specify better.
Lines 140-141: The phrase contains the repetition of year and amount of Gtons
Author Response
Point 1: The contents of the paper are interesting but the entire paper seems to be a sort of introduction to the matter. The simulation model is described but deeper details on possible applications are missing. The overall impression is of a very elementary dissertation, that needs a deeper and more extensive analysis to be published. Moreover, even the English language used to present them is quite elementary and confused.
Response 1: The article was edited in the parts of "Abstract", "Introduction", "Methods", "Results and Discussions", "Conclusions", "References"
Point 2: Line 115: maybe you intended “Energy production” not “Energy”.
Response 2: Edited.
Point 3: Lines 115-117: the phrase “At the same time …… consumption of energy” is not clear, please specify better.
Response 3: Edited.
Point 4: Lines 140-141: The phrase contains the repetition of year and amount of Gton.
Response 4: Edited.
Round 2
Reviewer 1 Report
I recommend to accept this manuscript in this form.
Author Response
Point 1: There are some typos in the manuscript and the English in the manuscript should be polished
Response 1: The English in the manuscript was polished. (in green)
Point 2: The smart transition to climate management used in this manuscript shoule be introduced in the part of "Induction", and some researched papers related to this method should also be added.
Response 2: The information about The smart transition to climate management was introduced in the parts of "Introduction" and "Methods" (in yellow and grey)
- Introduction:
“Introducing smart city and climate-neutral technologies in energy security are particularly essential [15-16]. In addition, the European Energy Efficiency Directive (EED 2012/2018) obliges the Member States to have all electricity meters remotely accessible for reading until January 2027 [18]. [19] proposed a plan for a provincial integrated energy services platform based on SCADA. The platform is based on CPS and uses the development of intelligent interactive and business management applications as the main line of communication with electricity consumers, energy service providers, government departments and other parties. In turn, using the EnergyPlan platform [20] makes it possible to simulate the operation of intelligent energy systems using renewable energy sources.
For reliable heat supply and cooling, the use of a geographic information system (GIS) identifies heat sources that can be used to provide heat or remove excess heat. [21] proposed a method for identifying possible heat sources for large heat pumps and chillers, combining geospatial data on administrative units, industrial facilities and natural water bodies.
In the context of the study of the peculiarities of the transition to climate management of the green energy transmission chain and the integration of smart technologies in the energy, the importance will be to use of smart approach to climate management of the green energy transmission chain based on intersectoral cooperation and the circular use of energy from renewable sources.”
- Methods
“Highlighting the trend of integrating smart technologies into the energy sector was the basis for studying the peculiarities of a smart transition to climate management of the green energy transmission chain. In this study, simulation modeling made it possible to develop an algorithm for applying a smart approach to climate management of the green energy transmission chain based on the work of Industry 4.0 technologies.
It should be noted that a distinctive feature of forming a climate-neutral green energy supply chain is the provision of a closed cycle of energy resources consumption (primary and secondary energy) and, as a result, the achievement of a zero-carbon footprint in climate change mitigation measures. In such a chain, we have proposed to single out the subjects of production and supply of primary energy (agricultural enterprises as producers of biomass, i.e., energy plants), processing of primary energy into secondary energy (enterprises to produce biofuels and enterprises of “green” thermal power industry), supply and service of secondary energy (distribution stations, energy service companies) directly to consumers (households, transport sector).
The innovativeness of building such a chain was in the observance of the climate neutrality principle at all stages of energy conversion, which involves laying the basis for the management model for the provision of “green” energy services to obtain such an optimization effect as maximizing the environmental effect (decarbonization of the environment) and minimizing the cost of energy consumption [24]:
(1)
(2)
where і – functions of maximizing the environmental effect from the provision of “green” energy services and minimizing the cost of energy consumption ; – an indicator of the decarbonization level for the use of “green” energy services ; – consumption costs of the corresponding type of energy ; n – the number of types of “green” energy services (i=1…n); m – the number of energy types (j=1…m).
Applying simulation modeling in Figure 1 showed an algorithm for a smart transition to climate management of the green energy transmission chain based on intersectoral interaction through a closed cycle of energy resources use, which involves the creation of regional climate-energy clusters.
Figure 1. Smart Approach to Climate Management of the Green Energy Transmission Chain.
1 Source: author’s development.
In order to optimize costs and increase the efficiency of the participants in such a chain (climatic energy cluster), the following functions are provided:
- smart industry centers for managing climate capacity (maybe at the level of one enterprise or group of enterprises in the industry) responsible for monitoring the economic and environmental efficiency of production, the use of energy resources (agricultural smart climate capacity management centre; bioprocessing (bioenergy, defense, food) smart -climatic capacity management centre, “green” energy (electricity/heat) climate capacity management smart centre, a consumer (household, transport, industrial) climate capacity management smart centre;
- intersectoral smart centers for climate interaction responsible for optimizing risk factors in the transfer of energy resources;
- energy service smart centers, which are responsible for supporting the adoption of energy-efficient decisions at the industry level (individual enterprise) and the introduction of climate-neutral technologies in the energy chain;
- smart reverse logistics centre, which is responsible for processing waste (reverse logistics) from the main activity of consumers of energy resources (for example, the use of waste heat from transport activities to produce thermal energy).
At the same time, the global community is currently underprepared for the growing intensity, frequency, and prevalence of climate change impacts, significantly as emissions rise. Climate resilience needs to be built quickly – moving from heightened public awareness and concern to massive adaptation action. Applying innovative measures for intersectoral interaction to enhance environmental and energy security and transition to climate-controlled energy resource management based on the circular economy and Industry 4.0 technologies is particularly important.”
Point 3: The results in the part 4 are not clearly and the authors are suggested to give more details, meanwhile, the datas in this part should be tested to credible.
Response 3:
- The part “Results and Discussions” was edited in the following part (in yellow):
At the same time, in order to improve the European integration processes in Ukraine and transnational cooperation in adapting to climate change in the energy sector, a transition to the digitalization of the energy infrastructure, the development of "smart" energy networks, and the promotion of environmental and energy literacy of the population are required. The urgency of solving this problem is complemented by the fact that by 2025 the reform of the energy complex of Ukraine should be completed, priority targets for security and energy efficiency should be achieved, and its innovative renewal and integration with the EU energy sector should be ensured. The solution to this complex problem is impossible without a profound transformation of the entire energy sector of the economy based on sustainable development and strengthening cooperation between Ukraine and the European Union. In particular, this implies the need to develop a strategy for the digital development of "green" energy; tools for building an innovative business model of "green" energy based on digital technologies, and methods for assessing the level of digital transformation of "green" energy, analyzing the structures of "green" energy by the criterion of the degree of digital transformation; organizational support for the digitalization of green energy structures.
Implementing these tasks will improve Ukraine's cross-border cooperation with the countries of the European Union on the application of an innovative approach to achieve optimal energy security and energy efficiency and the implementation of joint measures to adapt to climate change. In addition, this will contribute to the intensive attraction of investments in the renewable energy sector, the development of distributed generation, in particular, the development and implementation of a plan for the implementation of "smart" energy networks (Smart Grids) and the creation of extensive electric transport infrastructure in the cross-border zone.”
- The part “Conclusions” was edited in the following part (in yellow):
“This provision is based on the idea that the transition to climate neutrality of energy development by 2050 by the European Union and Ukraine provides the implementation of transnational measures to diversify the sources of obtaining and using "green" energy by enterprises and households, to optimize business processes for the production of "green" energy and to ensure integrated communication of "green" energy producers, local governments, energy service companies and consumers.
In this context, one should take into account the fact that the use of digital technologies in various areas of life contributes to the inclusive development of the economy, improving the level and quality of life of the population, the availability and rational use of resources, ensuring security, and automated processing of large databases. In addition, in the context of decentralization, the role of local governments as facilitators in forming a social network to ensure access to "green" energy through the transition to the smart specialization of the energy sector at the regional level is increasing. Today, economic growth, social justice and environmental protection are interrelated components of cyclic socio-economic development. The prospect of developing a unified system of models within the framework of cross-border and transnational cooperation will be aimed at optimizing the development of "green" energy through the digitalization of energy infrastructure, the development of "smart" energy networks, improving energy efficiency, and increasing energy and environmental security of the population.”
Point 4: Extensive editing of English language and style required. Are all the cited references relevant to the research?
Response 4:
- the English was edited (in green);
- the cited references was clarified in the part “References”(in grey and yellow):
References
- Delivering the European Green Deal. Available online: https://ec.europa.eu/info/strategy/priorities-2019-2024 (accessed on 25 June 2022).
- European State of the Climate. Available online: https://climate.copernicus.eu/esotc/2021 (accessed on 25 June 2022).
- Brych, V.; Borysiak, O.; Halysh, N.; Liakhovych, G.; Kupchak, V.; Vakun, Impact of International Climate Policy on the Supply Management of Enterprises Producing Green Energy. Lecture Notes in Networks and Systems 2023, 485, 649-661.
- Song, J.; Zhang, L.; Jiang, Q.; Ma, Y.; Zhang, X.; Xue, G.; Shen, X.; Wu, X. Estimate the daily consumption of natural gas in district heating system based on a hybrid seasonal decomposition and temporal convolutional network model. Applied Energy 2022, 309.
- Yin, L.; Qiu, Long-term price guidance mechanism of flexible energy service providers based on stochastic differential methods. Energy 2022, 238, 121818.
- Bashir, A.A.; Jokisalo, J.; Heljo, J.; Safdarian, A.; Lehtonen, M. Harnessing the Flexibility of District Heating System for Integrating Extensive Share of Renewable Energy Sources in Energy Systems. IEEE Access 2021, 9, 116407-116426.
- Halysh, N.; Borysiak, O.; Brych, V.; Korol, V.; Vakun, O.; Zaburanna, Technical and Economic Analysis of Implementation of Standards for Solid Fuels. Lecture Notes in Networks and Systems 2021, 194, 931–942.
- Dluhopolskyi, O.; Brych, V.; Borysiak, O.; Fedirko, M.; Dziubanovska, N.; Halysh, N. Modeling the Environmental and Economic Effect of Value Added Created in the Energy Service Market. Polityka Energetyczna 2021, 24(4), 153-164.
- Leiria, D.; Johra, H.; Marszal-Pomianowska, A.; Pomianowski, M.; Heiselberg, Using data from smart energy meters to gain knowledge about households connected to the district heating network: a Danish case. Smart Energy 2021, 3, 100035.
- Dombrovskyi, O.; Heletukha, H. Paris Climate Agreement: Ukraine must reduce emissions by 70%. Available online: https://www.epravda.com.ua/publications/2016/03/18/585855/ (accessed on 25 June 2022).
- Inaniuta, S.P.; Kolomiiets, O.O.; Malynovska, O.A.; Yakushenko, M. Climate change: consequences and adaptation measures. Analytical Report 2020. К.: NІSD.
- Zhang, Y.; Johansson, P.; Kalagasidis, A.S. Assessment of district heating and cooling systems transition with respect to future changes in demand profiles and renewable energy supplies. Energy Conversion and Management 2022, 268, 116038.
- Quirosa, G.; Torres, m.; Soltero, V.M.; Chacartegui, R. Analysis of an ultra-low temperature district heating and cooling as a storage system for renewable integration. Applied Thermal Engineering 2022, 216, 119052.
- Li, X.; Walch, A.; Yilmaz, S.; Patel, M.; Chambers, J. Optimal spatial resource allocation in networks: Application to district heating and cooling. Computers & Industrial Engineering 2022, 171, 108448.
- Silva, P.; Pires, S.M.; Teles, F.; Polido, A.; Rodrigues, C. Pre-conditions and barriers for territorial innovation through smart specialization strategies: the case of the lagging Centro region of Portugal. Urban Research & Practice
- Brych, V.; Manzhula, V.; Borysiak, O.; Liakhovych, G.; Halysh, N.; Tolubyak, V. Communication Model of Energy Service Market Participants in the Context of Cyclic Management City Infrastructure. 10th International Conference on Advanced Computer Information Technologies (ACIT) 2020, 678-681.
- Chiordi, S.; Desogus, G.; Garau, C.,; Nesi, P.; Zamperlin, P. A Preliminary Survey on Smart Specialization Platforms: Evaluation of European Best Practices. Lecture Notes in Computer Science 2022,13382.
- Wang, X.; Zhang, X.; Duan, J.; Chen, W.; Sun, X.; Xia, J. Design of Provincial Comprehensive Energy Service Platform Based on SCADA. Communications in Computer and Information Science 2022, 1587.
- Tunzi, M.; Svendsen, S. Digitalization of the Demand-Side: The enabler for low-temperature operations in existing buildings connected to district heating networks.Hot Cool 2022, 4, 7-9.
- Lund, H.; Thellufsen, J.Z.; Østergaard, P.A.; Sorknæs, P.; Skov, I.R.; Mathiesen, B.V. EnergyPLAN – Advanced analysis of smart energy systems. Smart Energy 2021, 1, 100007.
- Pieper, H.; Lepiksaar, K.; Volkova, A. GIS-based approach to identifying potential heat sources for heat pumps and chillers providing district heating and cooling. International Journal of Sustainable Energy Planning and Management 2022, 34, 29–44.
- Brych, V.; Borysiak, O.; Yushchenko, N.; Bondarchuk, M.; Alieksieiev, I.; Halysh, N. Factor Modeling of the Interaction of Agricultural Enterprises and Enterprises Producing Green Energy to Optimize the Biomass Supply Chain. 11th International Conference on Advanced Computer Information Technologies (ACIT) 2021, 424-427.
- Borysova, T.; Monastyrskyi, G.; Borysiak, O.; Protsyshyn, Y. Priorities of Marketing, Competitiveness, and Innovative Development of Transport Service Providers under Sustainable Urban Development. Marketing and Management of Innovations 2021, 3, 78-89.
- Borysiak, O.; Brych, Methodological Approach to Assessing the Management Model of Promoting Green Energy Services in the Context of Development Smart Energy Grids.Financial and credit activity: problems of theory and practice 2021, 4(39), 302-309.
- Brych, V.; Manzhula, V.; Borysiak, O.; Bondarchuk, M.; Alieksieiev, I.; Halysh, Factor Analysis of Financial and Economic Activities of Energy Enterprises of Ukraine. 11th International Conference on Advanced Computer Information Technologies (ACIT) 2021, 415-419.
- Shuvar, I.; Korpita, H.; Balkovskyi, V.; Shuvar, A. Peculiarities of yield formation of potato depending on the climate conditions of the western forest steppe of Ukraine. E3S Web of Conferences2021, 254, 02016.
- Shuvar, A.; Rudavska, N.; Shuvar, I.; Korpita, Realization of genetic potential of fiber flax varieties under the influence of growth stimulators of organic origin. E3S Web of Conferences2021, 254, 03004.
- Brych, V.; Kalinichuk, N.; Halysh, N.; Borysiak, O.; Shushpanov, D.; Zagurskyy, O. Dynamics of living standards based on factors of the remuneration system. Lecture Notes in Networks and Systems 2023, 487, 597-607.
- Borysiak, O.; Brych, V. Post-COVID-19 Revitalization and Prospects for Climate Neutral Energy Security Technologies. Problemy Ekorozwoju 2022, 17(2), 31-38.
- Development of renewable energy sources in Ukraine: Energy of Ukraine 2021. Infographic study on Ukraine’s energy. Available online: https://businessviews.com.ua/ru/get_file/id/energy-of-ukraine-2021.pdf (accessed on 25 June 2022).
- Global Energy Review 2020. The impacts of the Covid-19 crisis on global energy demand and CO2 emissions. International Energy Agency. Available online: https://www.iea.org/reports/global-energy-review-2020 (accessed on 25 June 2022).
- Brych, V.; Zatonatska, T.; Dluhopolskyi, O.; Borysiak, O.; Vakun, O. Estimating the efficiency of the green energy services’ marketing management based on segmentation. Marketing and Management of Innovations 2021, 3, 188-198.
- Sejkora, C.; Kühberger, L.; Radner, F.; Trattner, A.; Kienberger, T. Exergy as criteria for efficient energy systems – Maximising energy efficiency from resource to energy service, an Austrian case study. Energy 2022, 239, 122173.
- Popovych, P., Shevchuk, O.; Dzyura, V.; Poberezhna, L.; Dozorskyy, V.; Hrytsanchuk, A. Assessment of the influence of corrosive aggressive cargo transportation on vehicle reliability. International Journal of Engineering Research in Africa 2018, 38, 17-25.
- Polianovskyi, H.; Zatonatska, T., Dluhopolskyi, O.; Liutyi, I. Digital and technological support of distance learning at universities under COVID-19 (case of Ukraine). Revista Romaneasca pentru Educatie Multidimensionala 2021, 13(4), 595-613.
- Heller, A.; Rasmussen, E.L. HEAT 4.0 takes the district heating sector into the next digital level. Hot Cool 2022, 1, 16-19.
- Rymarczyk, T.; Król, K.; Kozłowski, E.; Wołowiec, T.; Cholewa-Wiktor, M.; Bednarczuk, P. Application of Electrical Tomography Imaging Using Machine Learning Methods for the Monitoring of Flood Embankments Leaks. Energies2021, 14, 8081.
Zhuravka, F.; Filatova H.; Šuleř P.; Wołowiec T. State debt assessment and forecasting: time series analysis. Investment Management and Financial Innovations

Reviewer 2 Report
Please, all suggestions need to be answered.
Author Response
Point 1: The Abstract should contain answers to the following questions: What problem was studied and why is it important? What are the important results? What conclusions can be drawn from the results? What is the novelty of the work and where does it go beyond previous efforts in the literature?
Response 1: The abstract was edited in the following parts:
- clarifying the importance of the problem (in grey and yellow):
“Climate challenges in recent decades have forced a change in attitude towards forms of environmental interaction. The International Climate Conference COP26 evidences the relevance of the climate issue at the global level in Glasgow (November 2021). The decrease in natural energy resources leads to searching for alternative energy sources. Given this, the article is devoted to studying the peculiarities of the transition to climate management of the green energy transmission chain based on the circular economy and smart technologies. The paper has used simulation modeling to develop an algorithm for applying a smart approach to climate management of the green energy transmission chain based on the work of Industry 4.0 technologies.”
- clarifying the important results, conclusions and the novelty of the work (in yellow):
“The result of this modeling will be the importance of strengthening the ability to develop intersectoral partnerships to create climate-energy clusters based on a closed cycle of using energy resources and developing smart technologies. At the same time, it has been found that COVID-19 has changed the behaviour of energy consumers towards the transition to the use of energy from renewable sources, at the same time carbon neutral. With this in mind, the article has assessed the climate capacity of industries to use green energy from renewable sources based on resource conservation (rational use of energy resources) and climate neutrality. The industries of Ukraine, which are the largest consumers of energy and, at the same time, are significantly affected by climate change, were taken for the study: industry, transport, and agriculture. The methodology for determining the indicator of the climate capacity of sectors in the transition to green energy has been based on the correlation index (ratio) of the consumption indicator of various types of energy by industries (petroleum products; natural gas; biofuels, and waste; electricity) and the indicator of gross value added of industries in pre-COVID-19 and COVID-19 conditions. The results have indicated that the use of energy from renewable sources (biofuels and waste) for the production of goods and services, as well as the economical nature of the provision of raw materials (biomass and faeces), is a factor ensures climate industry neutrality and enhances its climate capability. The prospects of such effects of assessing the climate capacity of sectors will be the basis for the rationale to develop intersectoral partnerships to create climate-energy clusters based on a closed cycle of using energy resources and developing smart technologies.”
Point 2: An updated and complete literature review should be conducted and should appear as part of the Introduction.
Point 4: In the introduction section, the literature studies should be rewritten and elaborated. Further information should be given about the literature studies and its obtained results
Response 2 and 4:
- the literature review was appeared in the part ‘’ Introduction’’;
- updated and complete literature review (in yellow):
“According to the International Energy Agency, energy efficiency (40%) and renewable energy (30%) will play a critical role in preventing global temperatures from rising more than 2°C and reducing CO2 emissions by 2050 [10]. Renewable energy is critical in decarbonizing the electrical system and mitigating the effects of anthropogenic climate change. However, renewable energy accounts for no more than 25% of the world’s generating capacity, with 16% hydropower and about 5% solar (SPP) and wind (WPP) power plants. Hydropower is vulnerable to changes in river water levels and temperature due to global warming [11].
It was predicted that district power supply systems will move to low-temperature district heating and high-temperature district cooling [12]. [13] investigated the application of ultra-low temperature district heating and cooling systems with operating temperatures from 6 to 40 °C to integrate renewable sources with a storage strategy using the distribution network as a storage system.
At the same time, it should be noted that district heating and cooling networks connect and distribute thermal energy resources in the network by sources and needs. As a result, ensuring the optimal distribution of thermal resources in a spatially distributed network and creating carbon-neutral energy systems play a unique role. [14] proposed a spatial clustering method, transportation theory, and linear programming to maximize distributed resources under spatial constraints, allowing large-scale analysis of a wide range of geospatially constrained resources, especially when applying renewable energy mapping to supply district heating and cooling.
Introducing smart city and climate-neutral technologies in energy security are particularly essential [15-16]. In addition, the European Energy Efficiency Directive (EED 2012/2018) obliges the Member States to have all electricity meters remotely accessible for reading until January 2027 [18]. [19] proposed a plan for a provincial integrated energy services platform based on SCADA. The platform is based on CPS and uses the development of intelligent interactive and business management applications as the main line of communication with electricity consumers, energy service providers, government departments and other parties. In turn, using the EnergyPlan platform [20] makes it possible to simulate the operation of intelligent energy systems using renewable energy sources.
For reliable heat supply and cooling, the use of a geographic information system (GIS) identifies heat sources that can be used to provide heat or remove excess heat. [21] proposed a method for identifying possible heat sources for large heat pumps and chillers, combining geospatial data on administrative units, industrial facilities and natural water bodies.”
Point 3: The summary part of this proposed study should be written more comprehensively. For example, which method was used, what are the parameters examined, it should contain statements containing the results obtained.
Response 3:
A detailed description of research methods is provided in the part “Methods” (in grey and yellow). . The paper has used simulation modeling to develop an algorithm for applying a smart approach to climate management of the green energy transmission chain based on the work of Industry 4.0 technologies. The methodology for determining the indicator of the climate capacity of sectors in the transition to green energy has been based on the correlation index (ratio) of the consumption indicator of various types of energy by industries (petroleum products; natural gas; biofuels, and waste; electricity) and the indicator of gross value added of industries in pre-COVID-19 and COVID-19 conditions.
“Combining sectors is necessary to effectively integrate renewable energy sources since almost all renewable energy sources depend on variations in environmental parameters [13]. In this context, an important role plays the development of an algorithm for applying the smart approach to climate management of the green energy transmission chain based on the circular interaction of enterprises in the agricultural, energy, and transport industries. The concept of the study is that the production and transition to the recycling of agro-bioresources is both a way to neutralize the negative impact on the climate (growing photosynthetic plants) and an alternative source of energy (biofuels). It has been established that in thermal power engineering, the use of renewable energy sources (biofuels) and waste energy obtained from the transport and housing and communal (household) sectors is regarded as a way to transition to climate-neutral thermal energy production. In particular, in thermal power plants, the transfer of such waste energy occurs based on reverse logistics (recycling of energy resources).
To achieve the established purpose and solve certain items, we used general scientific and specific methods, such as content analysis, system analysis, expert assessments, economic-statistical and comparative analysis, simulation modelling, graphical and tabular presentation, and abstract logical ways.
The idea of substantiation of transition to climate management of the green energy supply chain by clustering enterprises in the agricultural, energy, energy service, transport and housing and communal (household) sectors based on the circular economy is justified by a combination of systemic and synergistic approaches to achieve the goal of the research, which made it possible to determine peculiarities of the transition to climate management of the “green” energy transmission chain. The methodological novelty of this approach is the possibility of developing a unified system of models within the framework of intersectoral cooperation, aimed at optimizing the development of “green” agriculture and energy by optimizing the processes of production, supply, and consumption of plant bioresources (agro-raw materials), the transition to the basics of a circular economy, the safety of the population, and the development of “green” transport.
The hypothesis was formed based on the results of our previous studies to determine the factors of influence on the interaction of agricultural enterprises and enterprises for the production of “green” energy to optimize the biomass supply chain [22], modeling the communication algorithm of energy service companies, transport users in the transition to the “green” energy consumption [23], the use of optimization methods and models to study the benefits of the transition to “green” energy (maximizing the decarbonization of energy and minimizing the cost of energy consumption) [24-25].
Highlighting the trend of integrating smart technologies into the energy sector was the basis for studying the peculiarities of a smart transition to climate management of the green energy transmission chain. In this study, simulation modeling made it possible to develop an algorithm for applying a smart approach to climate management of the green energy transmission chain based on the work of Industry 4.0 technologies.
It should be noted that a distinctive feature of forming a climate-neutral green energy supply chain is the provision of a closed cycle of energy resources consumption (primary and secondary energy) and, as a result, the achievement of a zero-carbon footprint in climate change mitigation measures. In such a chain, we have proposed to single out the subjects of production and supply of primary energy (agricultural enterprises as producers of biomass, i.e., energy plants), processing of primary energy into secondary energy (enterprises to produce biofuels and enterprises of “green” thermal power industry), supply and service of secondary energy (distribution stations, energy service companies) directly to consumers (households, transport sector).
The innovativeness of building such a chain was in the observance of the climate neutrality principle at all stages of energy conversion, which involves laying the basis for the management model for the provision of “green” energy services to obtain such an optimization effect as maximizing the environmental effect (decarbonization of the environment) and minimizing the cost of energy consumption [24]:
(1)
(2)
where і – functions of maximizing the environmental effect from the provision of “green” energy services and minimizing the cost of energy consumption ; – an indicator of the decarbonization level for the use of “green” energy services ; – consumption costs of the corresponding type of energy ; n – the number of types of “green” energy services (i=1…n); m – the number of energy types (j=1…m).
Applying simulation modeling in Figure 1 showed an algorithm for a smart transition to climate management of the green energy transmission chain based on intersectoral interaction through a closed cycle of energy resources use, which involves the creation of regional climate-energy clusters.
Figure 1. Smart Approach to Climate Management of the Green Energy Transmission Chain.
1 Source: author’s development.
In order to optimize costs and increase the efficiency of the participants in such a chain (climatic energy cluster), the following functions are provided:
- smart industry centers for managing climate capacity (maybe at the level of one enterprise or group of enterprises in the industry) responsible for monitoring the economic and environmental efficiency of production, the use of energy resources (agricultural smart climate capacity management centre; bioprocessing (bioenergy, defense, food) smart -climatic capacity management centre, “green” energy (electricity/heat) climate capacity management smart centre, a consumer (household, transport, industrial) climate capacity management smart centre;
- intersectoral smart centers for climate interaction responsible for optimizing risk factors in the transfer of energy resources;
- energy service smart centers, which are responsible for supporting the adoption of energy-efficient decisions at the industry level (individual enterprise) and the introduction of climate-neutral technologies in the energy chain;
- smart reverse logistics centre, which is responsible for processing waste (reverse logistics) from the main activity of consumers of energy resources (for example, the use of waste heat from transport activities to produce thermal energy).
At the same time, the global community is currently underprepared for the growing intensity, frequency, and prevalence of climate change impacts, significantly as emissions rise. Climate resilience needs to be built quickly – moving from heightened public awareness and concern to massive adaptation action. Applying innovative measures for intersectoral interaction to enhance environmental and energy security and transition to climate-controlled energy resource management based on the circular economy and Industry 4.0 technologies is particularly important.
Given this, we have proposed to assess the climate capacity of industries that are the largest consumers of energy and, at the same time, are affected by climate change transition (for example, industry, transport, and agriculture in Ukraine) to green energy based on resource conservation (rational use of energy resources) and climate neutrality [26–28]. At the same time, it should be noted that COVID-19 [29] has changed the behaviour of energy consumers towards the transition to the use of energy from renewable sources, which is carbon neutral.
In the value structure of final products (output), energy use costs are classified as intermediate costs, while gross value added is defined as the difference between output and intermediate consumption. As a result, we have based the methodology for determining the indicator of the climate capacity of industries in the transition to green energy ( ) on the correlation index (ratio) of the indicator of consumption of various types of energy (petroleum products; natural gas; biofuels and waste; electricity) by industries () and the indicator of the gross additive of industries ( ) in pre-COVID-19 and COVID-19 conditions (Table 1):
where is the index of climate ability of transition to green energy i, c,…n+1- branches in pre-COVID-19 and COVID-19 conditions.;
is the value of the gross value added indicator of the i-sector (industry; transport; agriculture);
is the value of the indicator of energy consumption by its types (petroleum products; natural gas; biofuels and waste; electricity) i-sectors;
is the value of the indicator of energy consumption from renewable sources (biofuel and waste, electricity) of the i-sector.
Table 1. The indicator of the climate capacity of Ukrainian industries in the transition to green energy in pre-COVID-19 and COVID-19 conditions
|
Industry |
Transport |
Agriculture |
2016 |
5,8 |
2,2 |
0,1 |
2017 |
4,4 |
2,3 |
0,2 |
2018 |
2,1 |
1,5 |
0,2 |
2019 |
5,4 |
3,3 |
0,2 |
Source: author’s development
It was diagnosed that the use of natural energy sources is a factor that leads to an increase in the consumer cost of the finished product (service), which is accompanied by an increase in the energy intensity of the industry and leads to a decrease in gross value added. The use of such types of energy as biofuels and waste for the production of goods and services, given the economical nature of the provision of raw materials (biomass and waste), is a factor ensures climate industry neutrality and enhances its climate capability. Such results of the empirical study are the basis for substantiating the feasibility of justifying the development of intersectoral partnerships to create climate-energy clusters based on a closed cycle of using energy resources and the development of smart technologies.

Reviewer 3 Report
The Authors practically did not respond to my comments, nor did they make the recommended corrections and additions. The additions introduced by the Authors are of "cosmetic" character. This material still contains known information given in a general manner. I still do not know how the algorithm of "climate management in the green energy transmission chain" was developed and what the form of this algorithm is. Fig. 1 cannot be considered a developed algorithm, and the Authors did not present the mutual relations between the elements of this Fig. And the necessary and exhaustive commentary. As a philosopher of nature, I cannot imagine how to manage the climate. It is only possible to organize technical, technological and social activities limiting the effects of climate impacts. In summary, this material is a general collection of known information.
Author Response
Point 1: The uploaded material is chaotic and very vague. The content of the article does not indicate the legitimacy of developing the algorithm or the methodology of its development. Practically, the article does not contain an unambiguous algorithm that is to be the result of the Authors' analytical work. Fig. 1 is an illustrative drawing, not described in the content of the article and very little contributing and understandable. The validity of presenting the data contained in Table 1 concerning Ukraine and the forecasts presented in Table 2 in the context of the article has not been proved either.
Point 3: The current state of knowledge in the field of the discussed problem based on bibliographic sources. Methodology of the conducted research and analytical process, taking into account all boundary conditions. The developed algorithm in its full form, based on the presented methodology.
Response 1 and 3: The article was edited in the parts of "Abstract", "Introduction", "Methods", "Results and Discussions", "Conclusions", "References" (in grey, yellow):
- the uploaded material is structured according to the purpose;
- The current state of knowledge in the field of the discussed problem was edited in the part "Introduction" and "References". (in yellow)
According to the International Energy Agency, energy efficiency (40%) and renewable energy (30%) will play a critical role in preventing global temperatures from rising more than 2°C and reducing CO2 emissions by 2050 [10]. Renewable energy is critical in decarbonizing the electrical system and mitigating the effects of anthropogenic climate change. However, renewable energy accounts for no more than 25% of the world’s generating capacity, with 16% hydropower and about 5% solar (SPP) and wind (WPP) power plants. Hydropower is vulnerable to changes in river water levels and temperature due to global warming [11].
It was predicted that district power supply systems will move to low-temperature district heating and high-temperature district cooling [12]. [13] investigated the application of ultra-low temperature district heating and cooling systems with operating temperatures from 6 to 40 °C to integrate renewable sources with a storage strategy using the distribution network as a storage system.
At the same time, it should be noted that district heating and cooling networks connect and distribute thermal energy resources in the network by sources and needs. As a result, ensuring the optimal distribution of thermal resources in a spatially distributed network and creating carbon-neutral energy systems play a unique role. [14] proposed a spatial clustering method, transportation theory, and linear programming to maximize distributed resources under spatial constraints, allowing large-scale analysis of a wide range of geospatially constrained resources, especially when applying renewable energy mapping to supply district heating and cooling.
Introducing smart city and climate-neutral technologies in energy security are particularly essential [15-16]. In addition, the European Energy Efficiency Directive (EED 2012/2018) obliges the Member States to have all electricity meters remotely accessible for reading until January 2027 [18]. [19] proposed a plan for a provincial integrated energy services platform based on SCADA. The platform is based on CPS and uses the development of intelligent interactive and business management applications as the main line of communication with electricity consumers, energy service providers, government departments and other parties. In turn, using the EnergyPlan platform [20] makes it possible to simulate the operation of intelligent energy systems using renewable energy sources.
For reliable heat supply and cooling, the use of a geographic information system (GIS) identifies heat sources that can be used to provide heat or remove excess heat. [21] proposed a method for identifying possible heat sources for large heat pumps and chillers, combining geospatial data on administrative units, industrial facilities and natural water bodies.
In the context of the study of the peculiarities of the transition to climate management of the green energy transmission chain and the integration of smart technologies in the energy, the importance will be to use of smart approach to climate management of the green energy transmission chain based on intersectoral cooperation and the circular use of energy from renewable sources.
- the algorithm and the methodology of its research was edited in the part "Methods":
“Combining sectors is necessary to effectively integrate renewable energy sources since almost all renewable energy sources depend on variations in environmental parameters [13]. In this context, an important role plays the development of an algorithm for applying the smart approach to climate management of the green energy transmission chain based on the circular interaction of enterprises in the agricultural, energy, and transport industries. The concept of the study is that the production and transition to the recycling of agro-bioresources is both a way to neutralize the negative impact on the climate (growing photosynthetic plants) and an alternative source of energy (biofuels). It has been established that in thermal power engineering, the use of renewable energy sources (biofuels) and waste energy obtained from the transport and housing and communal (household) sectors is regarded as a way to transition to climate-neutral thermal energy production. In particular, in thermal power plants, the transfer of such waste energy occurs based on reverse logistics (recycling of energy resources).
To achieve the established purpose and solve certain items, we used general scientific and specific methods, such as content analysis, system analysis, expert assessments, economic-statistical and comparative analysis, simulation modelling, graphical and tabular presentation, and abstract logical ways.
The idea of substantiation of transition to climate management of the green energy supply chain by clustering enterprises in the agricultural, energy, energy service, transport and housing and communal (household) sectors based on the circular economy is justified by a combination of systemic and synergistic approaches to achieve the goal of the research, which made it possible to determine peculiarities of the transition to climate management of the “green” energy transmission chain. The methodological novelty of this approach is the possibility of developing a unified system of models within the framework of intersectoral cooperation, aimed at optimizing the development of “green” agriculture and energy by optimizing the processes of production, supply, and consumption of plant bioresources (agro-raw materials), the transition to the basics of a circular economy, the safety of the population, and the development of “green” transport.
The hypothesis was formed based on the results of our previous studies to determine the factors of influence on the interaction of agricultural enterprises and enterprises for the production of “green” energy to optimize the biomass supply chain [22], modeling the communication algorithm of energy service companies, transport users in the transition to the “green” energy consumption [23], the use of optimization methods and models to study the benefits of the transition to “green” energy (maximizing the decarbonization of energy and minimizing the cost of energy consumption) [24-25].
Highlighting the trend of integrating smart technologies into the energy sector was the basis for studying the peculiarities of a smart transition to climate management of the green energy transmission chain. In this study, simulation modeling made it possible to develop an algorithm for applying a smart approach to climate management of the green energy transmission chain based on the work of Industry 4.0 technologies.
It should be noted that a distinctive feature of forming a climate-neutral green energy supply chain is the provision of a closed cycle of energy resources consumption (primary and secondary energy) and, as a result, the achievement of a zero-carbon footprint in climate change mitigation measures. In such a chain, we have proposed to single out the subjects of production and supply of primary energy (agricultural enterprises as producers of biomass, i.e., energy plants), processing of primary energy into secondary energy (enterprises to produce biofuels and enterprises of “green” thermal power industry), supply and service of secondary energy (distribution stations, energy service companies) directly to consumers (households, transport sector).
The innovativeness of building such a chain was in the observance of the climate neutrality principle at all stages of energy conversion, which involves laying the basis for the management model for the provision of “green” energy services to obtain such an optimization effect as maximizing the environmental effect (decarbonization of the environment) and minimizing the cost of energy consumption [24]:
(1)
- (2)
where і – functions of maximizing the environmental effect from the provision of “green” energy services and minimizing the cost of energy consumption ; – an indicator of the decarbonization level for the use of “green” energy services ; – consumption costs of the corresponding type of energy ; n – the number of types of “green” energy services (i=1…n); m – the number of energy types (j=1…m).
Applying simulation modeling in Figure 1 showed an algorithm for a smart transition to climate management of the green energy transmission chain based on intersectoral interaction through a closed cycle of energy resources use, which involves the creation of regional climate-energy clusters.
Figure 1. Smart Approach to Climate Management of the Green Energy Transmission Chain.
1 Source: author’s development.
In order to optimize costs and increase the efficiency of the participants in such a chain (climatic energy cluster), the following functions are provided:
- smart industry centers for managing climate capacity (maybe at the level of one enterprise or group of enterprises in the industry) responsible for monitoring the economic and environmental efficiency of production, the use of energy resources (agricultural smart climate capacity management centre; bioprocessing (bioenergy, defense, food) smart -climatic capacity management centre, “green” energy (electricity/heat) climate capacity management smart centre, a consumer (household, transport, industrial) climate capacity management smart centre;
- intersectoral smart centers for climate interaction responsible for optimizing risk factors in the transfer of energy resources;
- energy service smart centers, which are responsible for supporting the adoption of energy-efficient decisions at the industry level (individual enterprise) and the introduction of climate-neutral technologies in the energy chain;
- smart reverse logistics centre, which is responsible for processing waste (reverse logistics) from the main activity of consumers of energy resources (for example, the use of waste heat from transport activities to produce thermal energy).
At the same time, the global community is currently underprepared for the growing intensity, frequency, and prevalence of climate change impacts, significantly as emissions rise. Climate resilience needs to be built quickly – moving from heightened public awareness and concern to massive adaptation action. Applying innovative measures for intersectoral interaction to enhance environmental and energy security and transition to climate-controlled energy resource management based on the circular economy and Industry 4.0 technologies is particularly important.
Given this, we have proposed to assess the climate capacity of industries that are the largest consumers of energy and, at the same time, are affected by climate change transition (for example, industry, transport, and agriculture in Ukraine) to green energy based on resource conservation (rational use of energy resources) and climate neutrality [26–28]. At the same time, it should be noted that COVID-19 [29] has changed the behaviour of energy consumers towards the transition to the use of energy from renewable sources, which is carbon neutral.
In the value structure of final products (output), energy use costs are classified as intermediate costs, while gross value added is defined as the difference between output and intermediate consumption. As a result, we have based the methodology for determining the indicator of the climate capacity of industries in the transition to green energy ( ) on the correlation index (ratio) of the indicator of consumption of various types of energy (petroleum products; natural gas; biofuels and waste; electricity) by industries () and the indicator of the gross additive of industries ( ) in pre-COVID-19 and COVID-19 conditions (Table 1):
where is the index of climate ability of transition to green energy i, c,…n+1- branches in pre-COVID-19 and COVID-19 conditions.;
- is the value of the gross value added indicator of the i-sector (industry; transport; agriculture);
- is the value of the indicator of energy consumption by its types (petroleum products; natural gas; biofuels and waste; electricity) i-sectors;
- is the value of the indicator of energy consumption from renewable sources (biofuel and waste, electricity) of the i-
Table 1. The indicator of the climate capacity of Ukrainian industries in the transition to green energy in pre-COVID-19 and COVID-19 conditions
|
Industry |
Transport |
Agriculture |
2016 |
5,8 |
2,2 |
0,1 |
2017 |
4,4 |
2,3 |
0,2 |
2018 |
2,1 |
1,5 |
0,2 |
2019 |
5,4 |
3,3 |
0,2 |
Source: author’s development
It was diagnosed that the use of natural energy sources is a factor that leads to an increase in the consumer cost of the finished product (service), which is accompanied by an increase in the energy intensity of the industry and leads to a decrease in gross value added. The use of such types of energy as biofuels and waste for the production of goods and services, given the economical nature of the provision of raw materials (biomass and waste), is a factor ensures climate industry neutrality and enhances its climate capability. Such results of the empirical study are the basis for substantiating the feasibility of justifying the development of intersectoral partnerships to create climate-energy clusters based on a closed cycle of using energy resources and the development of smart technologies.”
Point 2: A research (analytical) problem with an indication and justification of the need to solve this problem.
Response 2: A research (analytical) problem with an indication and justification was edited in the part "Introduction". (in grey and yellow)
The consequences of climate change lead to the search for innovative approaches to the economical use of natural resources by strengthening energy security. The current European action is to develop measures to adapt to climate change by 2030 and 2050 and disseminate them through the formation of an extensive regulatory framework (for example, the European Green Deal (2019) [1], Forging a climate-resilient Europe Climate Change (2021) [2]). Their principles are implemented by developing political and legal instruments of the national climate policy (Nationally determined contributions, Long-term strategies for low-carbon development, National plans for energy and climate, and National strategy for adaptation to climate change). The EU is committed to climate neutrality by 2050 and has a more ambitious goal of reducing emissions by at least 55% by 2030 compared to 1990.
The energy sector is one of the industries whose activities negatively impact the climate [3]. In recent years, natural gas has been widely used as a primary clean energy source to replace coal, aiming to reduce the severe environmental pollution caused by coal-fired district heating in winter [4]. The rapid growth in natural gas consumption has placed a significant strain on natural gas production and transportation, which has affected residents’ regular demand for heating. Therefore, a district heating (DH) system must accurately forecast natural gas consumption. In [5], a long-term price guidance mechanism for flexible energy service providers is considered which based on stochastic differential methods that mobilize energy flexibility by indirectly controlling demand for flexible energy systems using reasonable price signals.
Recently, the European Union has tightened the targets set to reduce carbon emissions. The energy production sector, particularly the district heating system, is still dominated by the combustion of fossil fuels, which significantly contributes to such emissions [6-7]. At the same time, one of the most sustainable solutions for heating buildings in Europe is district heating, highlighting the need to integrate renewable energy sources into the heating supply. While district heating is vital to a sustainable future, it requires extensive planning and long-term investment [8-9].
According to the International Energy Agency, energy efficiency (40%) and renewable energy (30%) will play a critical role in preventing global temperatures from rising more than 2°C and reducing CO2 emissions by 2050 [10]. Renewable energy is critical in decarbonizing the electrical system and mitigating the effects of anthropogenic climate change. However, renewable energy accounts for no more than 25% of the world’s generating capacity, with 16% hydropower and about 5% solar (SPP) and wind (WPP) power plants. Hydropower is vulnerable to changes in river water levels and temperature due to global warming [11].
It was predicted that district power supply systems will move to low-temperature district heating and high-temperature district cooling [12]. [13] investigated the application of ultra-low temperature district heating and cooling systems with operating temperatures from 6 to 40 °C to integrate renewable sources with a storage strategy using the distribution network as a storage system.
At the same time, it should be noted that district heating and cooling networks connect and distribute thermal energy resources in the network by sources and needs. As a result, ensuring the optimal distribution of thermal resources in a spatially distributed network and creating carbon-neutral energy systems play a unique role. [14] proposed a spatial clustering method, transportation theory, and linear programming to maximize distributed resources under spatial constraints, allowing large-scale analysis of a wide range of geospatially constrained resources, especially when applying renewable energy mapping to supply district heating and cooling.
Introducing smart city and climate-neutral technologies in energy security are particularly essential [15-16]. In addition, the European Energy Efficiency Directive (EED 2012/2018) obliges the Member States to have all electricity meters remotely accessible for reading until January 2027 [18]. [19] proposed a plan for a provincial integrated energy services platform based on SCADA. The platform is based on CPS and uses the development of intelligent interactive and business management applications as the main line of communication with electricity consumers, energy service providers, government departments and other parties. In turn, using the EnergyPlan platform [20] makes it possible to simulate the operation of intelligent energy systems using renewable energy sources.
For reliable heat supply and cooling, the use of a geographic information system (GIS) identifies heat sources that can be used to provide heat or remove excess heat. [21] proposed a method for identifying possible heat sources for large heat pumps and chillers, combining geospatial data on administrative units, industrial facilities and natural water bodies.
In the context of the study of the peculiarities of the transition to climate management of the green energy transmission chain and the integration of smart technologies in the energy, the importance will be to use of smart approach to climate management of the green energy transmission chain based on intersectoral cooperation and the circular use of energy from renewable sources.
Point 4: Specific and unambiguous conclusions presenting the practical possibilities of using the developed algorithm in environmental as well as technical, economic and social aspects. Major improvement needed to rework the article.
Response 4: The part “Conclusions” was edited in the following part (in yellow):
“This provision is based on the idea that the transition to climate neutrality of energy development by 2050 by the European Union and Ukraine provides the implementation of transnational measures to diversify the sources of obtaining and using "green" energy by enterprises and households, to optimize business processes for the production of "green" energy and to ensure integrated communication of "green" energy producers, local governments, energy service companies and consumers.
In this context, one should take into account the fact that the use of digital technologies in various areas of life contributes to the inclusive development of the economy, improving the level and quality of life of the population, the availability and rational use of resources, ensuring security, and automated processing of large databases. In addition, in the context of decentralization, the role of local governments as facilitators in forming a social network to ensure access to "green" energy through the transition to the smart specialization of the energy sector at the regional level is increasing. Today, economic growth, social justice and environmental protection are interrelated components of cyclic socio-economic development. The prospect of developing a unified system of models within the framework of cross-border and transnational cooperation will be aimed at optimizing the development of "green" energy through the digitalization of energy infrastructure, the development of "smart" energy networks, improving energy efficiency, and increasing energy and environmental security of the population.”

Reviewer 4 Report
The authors extensively revised the article, which is now ready for publication.
Author Response
Point 1: The contents of the paper are interesting but the entire paper seems to be a sort of introduction to the matter. The simulation model is described but deeper details on possible applications are missing. The overall impression is of a very elementary dissertation, that needs a deeper and more extensive analysis to be published. Moreover, even the English language used to present them is quite elementary and confused.
Response 1: The article was edited in the parts of "Abstract", "Introduction", "Methods", "Results and Discussions", "Conclusions", "References" (in grey, yellow):
- the uploaded material is structured according to the purpose;
- The current state of knowledge in the field of the discussed problem was edited in the part "Introduction" and "References". (in yellow)
According to the International Energy Agency, energy efficiency (40%) and renewable energy (30%) will play a critical role in preventing global temperatures from rising more than 2°C and reducing CO2 emissions by 2050 [10]. Renewable energy is critical in decarbonizing the electrical system and mitigating the effects of anthropogenic climate change. However, renewable energy accounts for no more than 25% of the world’s generating capacity, with 16% hydropower and about 5% solar (SPP) and wind (WPP) power plants. Hydropower is vulnerable to changes in river water levels and temperature due to global warming [11].
It was predicted that district power supply systems will move to low-temperature district heating and high-temperature district cooling [12]. [13] investigated the application of ultra-low temperature district heating and cooling systems with operating temperatures from 6 to 40 °C to integrate renewable sources with a storage strategy using the distribution network as a storage system.
At the same time, it should be noted that district heating and cooling networks connect and distribute thermal energy resources in the network by sources and needs. As a result, ensuring the optimal distribution of thermal resources in a spatially distributed network and creating carbon-neutral energy systems play a unique role. [14] proposed a spatial clustering method, transportation theory, and linear programming to maximize distributed resources under spatial constraints, allowing large-scale analysis of a wide range of geospatially constrained resources, especially when applying renewable energy mapping to supply district heating and cooling.
Introducing smart city and climate-neutral technologies in energy security are particularly essential [15-16]. In addition, the European Energy Efficiency Directive (EED 2012/2018) obliges the Member States to have all electricity meters remotely accessible for reading until January 2027 [18]. [19] proposed a plan for a provincial integrated energy services platform based on SCADA. The platform is based on CPS and uses the development of intelligent interactive and business management applications as the main line of communication with electricity consumers, energy service providers, government departments and other parties. In turn, using the EnergyPlan platform [20] makes it possible to simulate the operation of intelligent energy systems using renewable energy sources.
For reliable heat supply and cooling, the use of a geographic information system (GIS) identifies heat sources that can be used to provide heat or remove excess heat. [21] proposed a method for identifying possible heat sources for large heat pumps and chillers, combining geospatial data on administrative units, industrial facilities and natural water bodies.
In the context of the study of the peculiarities of the transition to climate management of the green energy transmission chain and the integration of smart technologies in the energy, the importance will be to use of smart approach to climate management of the green energy transmission chain based on intersectoral cooperation and the circular use of energy from renewable sources.
- the algorithm and the methodology of its research was edited in the part "Methods":
“Combining sectors is necessary to effectively integrate renewable energy sources since almost all renewable energy sources depend on variations in environmental parameters [13]. In this context, an important role plays the development of an algorithm for applying the smart approach to climate management of the green energy transmission chain based on the circular interaction of enterprises in the agricultural, energy, and transport industries. The concept of the study is that the production and transition to the recycling of agro-bioresources is both a way to neutralize the negative impact on the climate (growing photosynthetic plants) and an alternative source of energy (biofuels). It has been established that in thermal power engineering, the use of renewable energy sources (biofuels) and waste energy obtained from the transport and housing and communal (household) sectors is regarded as a way to transition to climate-neutral thermal energy production. In particular, in thermal power plants, the transfer of such waste energy occurs based on reverse logistics (recycling of energy resources).
To achieve the established purpose and solve certain items, we used general scientific and specific methods, such as content analysis, system analysis, expert assessments, economic-statistical and comparative analysis, simulation modelling, graphical and tabular presentation, and abstract logical ways.
The idea of substantiation of transition to climate management of the green energy supply chain by clustering enterprises in the agricultural, energy, energy service, transport and housing and communal (household) sectors based on the circular economy is justified by a combination of systemic and synergistic approaches to achieve the goal of the research, which made it possible to determine peculiarities of the transition to climate management of the “green” energy transmission chain. The methodological novelty of this approach is the possibility of developing a unified system of models within the framework of intersectoral cooperation, aimed at optimizing the development of “green” agriculture and energy by optimizing the processes of production, supply, and consumption of plant bioresources (agro-raw materials), the transition to the basics of a circular economy, the safety of the population, and the development of “green” transport.
The hypothesis was formed based on the results of our previous studies to determine the factors of influence on the interaction of agricultural enterprises and enterprises for the production of “green” energy to optimize the biomass supply chain [22], modeling the communication algorithm of energy service companies, transport users in the transition to the “green” energy consumption [23], the use of optimization methods and models to study the benefits of the transition to “green” energy (maximizing the decarbonization of energy and minimizing the cost of energy consumption) [24-25].
Highlighting the trend of integrating smart technologies into the energy sector was the basis for studying the peculiarities of a smart transition to climate management of the green energy transmission chain. In this study, simulation modeling made it possible to develop an algorithm for applying a smart approach to climate management of the green energy transmission chain based on the work of Industry 4.0 technologies.
It should be noted that a distinctive feature of forming a climate-neutral green energy supply chain is the provision of a closed cycle of energy resources consumption (primary and secondary energy) and, as a result, the achievement of a zero-carbon footprint in climate change mitigation measures. In such a chain, we have proposed to single out the subjects of production and supply of primary energy (agricultural enterprises as producers of biomass, i.e., energy plants), processing of primary energy into secondary energy (enterprises to produce biofuels and enterprises of “green” thermal power industry), supply and service of secondary energy (distribution stations, energy service companies) directly to consumers (households, transport sector).
The innovativeness of building such a chain was in the observance of the climate neutrality principle at all stages of energy conversion, which involves laying the basis for the management model for the provision of “green” energy services to obtain such an optimization effect as maximizing the environmental effect (decarbonization of the environment) and minimizing the cost of energy consumption [24]:
(1)
- (2)
where і – functions of maximizing the environmental effect from the provision of “green” energy services and minimizing the cost of energy consumption ; – an indicator of the decarbonization level for the use of “green” energy services ; – consumption costs of the corresponding type of energy ; n – the number of types of “green” energy services (i=1…n); m – the number of energy types (j=1…m).
Applying simulation modeling in Figure 1 showed an algorithm for a smart transition to climate management of the green energy transmission chain based on intersectoral interaction through a closed cycle of energy resources use, which involves the creation of regional climate-energy clusters.
Figure 1. Smart Approach to Climate Management of the Green Energy Transmission Chain.
1 Source: author’s development.
In order to optimize costs and increase the efficiency of the participants in such a chain (climatic energy cluster), the following functions are provided:
- smart industry centers for managing climate capacity (maybe at the level of one enterprise or group of enterprises in the industry) responsible for monitoring the economic and environmental efficiency of production, the use of energy resources (agricultural smart climate capacity management centre; bioprocessing (bioenergy, defense, food) smart -climatic capacity management centre, “green” energy (electricity/heat) climate capacity management smart centre, a consumer (household, transport, industrial) climate capacity management smart centre;
- intersectoral smart centers for climate interaction responsible for optimizing risk factors in the transfer of energy resources;
- energy service smart centers, which are responsible for supporting the adoption of energy-efficient decisions at the industry level (individual enterprise) and the introduction of climate-neutral technologies in the energy chain;
- smart reverse logistics centre, which is responsible for processing waste (reverse logistics) from the main activity of consumers of energy resources (for example, the use of waste heat from transport activities to produce thermal energy).
At the same time, the global community is currently underprepared for the growing intensity, frequency, and prevalence of climate change impacts, significantly as emissions rise. Climate resilience needs to be built quickly – moving from heightened public awareness and concern to massive adaptation action. Applying innovative measures for intersectoral interaction to enhance environmental and energy security and transition to climate-controlled energy resource management based on the circular economy and Industry 4.0 technologies is particularly important.
Given this, we have proposed to assess the climate capacity of industries that are the largest consumers of energy and, at the same time, are affected by climate change transition (for example, industry, transport, and agriculture in Ukraine) to green energy based on resource conservation (rational use of energy resources) and climate neutrality [26–28]. At the same time, it should be noted that COVID-19 [29] has changed the behaviour of energy consumers towards the transition to the use of energy from renewable sources, which is carbon neutral.
In the value structure of final products (output), energy use costs are classified as intermediate costs, while gross value added is defined as the difference between output and intermediate consumption. As a result, we have based the methodology for determining the indicator of the climate capacity of industries in the transition to green energy ( ) on the correlation index (ratio) of the indicator of consumption of various types of energy (petroleum products; natural gas; biofuels and waste; electricity) by industries () and the indicator of the gross additive of industries ( ) in pre-COVID-19 and COVID-19 conditions (Table 1):
where is the index of climate ability of transition to green energy i, c,…n+1- branches in pre-COVID-19 and COVID-19 conditions.;
- is the value of the gross value added indicator of the i-sector (industry; transport; agriculture);
- is the value of the indicator of energy consumption by its types (petroleum products; natural gas; biofuels and waste; electricity) i-sectors;
- is the value of the indicator of energy consumption from renewable sources (biofuel and waste, electricity) of the i-
Table 1. The indicator of the climate capacity of Ukrainian industries in the transition to green energy in pre-COVID-19 and COVID-19 conditions
|
Industry |
Transport |
Agriculture |
2016 |
5,8 |
2,2 |
0,1 |
2017 |
4,4 |
2,3 |
0,2 |
2018 |
2,1 |
1,5 |
0,2 |
2019 |
5,4 |
3,3 |
0,2 |
Source: author’s development
It was diagnosed that the use of natural energy sources is a factor that leads to an increase in the consumer cost of the finished product (service), which is accompanied by an increase in the energy intensity of the industry and leads to a decrease in gross value added. The use of such types of energy as biofuels and waste for the production of goods and services, given the economical nature of the provision of raw materials (biomass and waste), is a factor ensures climate industry neutrality and enhances its climate capability. Such results of the empirical study are the basis for substantiating the feasibility of justifying the development of intersectoral partnerships to create climate-energy clusters based on a closed cycle of using energy resources and the development of smart technologies.”
Point 2: Line 115: maybe you intended “Energy production” not “Energy”.
Response 2: Edited (in green):
“Energy production is one of the sources of greenhouse gas emissions.”
Point 3: Lines 115-117: the phrase “At the same time …… consumption of energy” is not clear, please specify better.
Response 3: Edited (in green):
“At the same time, seasonal decrease or increase in demand for energy resources at the consumer level due to climate change negatively affects the balancing of the energy system.”
Point 4: Lines 140-141: The phrase contains the repetition of year and amount of Gton.
Response 4: Edited (in green):
In 2020, an overall reduction of 2.58 Gt of energy-related CO2 emissions was recorded.
